# Rod-Shaped Starch from Galanga: Physicochemical Properties, Fine Structure and In Vitro Digestibility

**DOI:** 10.3390/foods13111784

**Published:** 2024-06-06

**Authors:** Shanshan Li, Rui He, Jiaqi Liu, Ying Chen, Tao Yang, Kun Pan

**Affiliations:** 1Hainan Provincial Key Laboratory for Research and Development of Tropical Herbs, School of Pharmacy, Hainan Medical University, Haikou 571199, China; lishanshan9797@163.com (S.L.); hr0823abc@163.com (R.H.); jiaqiliu49101@163.com (J.L.); taoyang@hainmc.edu.cn (T.Y.); 2School of Food Science and Engineering, Yangzhou University, Yangzhou 225127, China

**Keywords:** rod-shaped non-conventional starch, rhizome starch, *Galanga*, fine structures

## Abstract

This work investigated the physicochemical properties, structural characteristics, and digestive properties of two non-conventional starches extracted from Galanga: *Alpinia officinarum* Hance starch (AOS) and *Alpinia galanga* Willd starch (AGS). The extraction rates of the two starches were 22.10 wt% and 15.73 wt%, which is lower than widely studied ginger (*Zingiber officinale*, ZOS). But they contained similar amounts of basic constituents. AOS and AGS showed a smooth, elongated shape, while ZOS was an oval sheet shape. AOS and ZOS were C-type starches, and AGS was an A-type starch. AOS showed the highest crystallinity (35.26 ± 1.02%) among the three starches, possessed a higher content of amylose (24.14 ± 0.73%) and a longer amylose average chain length (1419.38 ± 31.28) than AGS. AGS starch exhibits the highest viscosity at all stages, while AOS starch shows the lowest pasting temperature, and ZOS starch, due to its high amylose content, displays lower peak and trough viscosities. Significant differences were also found in the physicochemical properties of the three starches, including the swelling power, solubility, thermal properties, and rheological properties of the three starches. The total content of resistant starch (RS) and slowly digestible starch (SDS) in AOS (81.05%), AGS (81.46%), and ZOS (82.58%) are considered desirable. These findings proved to be valuable references for further research and utilization of ginger family starch.

## 1. Introduction

Starch, the main carbohydrate reserve found in plant roots and tubers [1], is widely used in food [2], pharmaceutical [3], and textile [4] industries thanks to its economic viability, widespread availability, and biological safety. The conventional sources of starch are corn, potato, wheat, rice, and cassava [5]. The physicochemical properties, gelation properties, and physiological functions of starch granules differ with the source of starch [6]. The utilization of starch is predominantly dictated by its physicochemical characteristics, encompassing moisture level, carbohydrate, and amylose content, swelling capacity, solubility, and viscosity [6]. Consequently, researchers are seeking new sources of starch from non-conventional crops and evaluating their physicochemical properties to determine their industrial potential.

Zingiberaceae, commonly known as the ginger family, is divided into about 53 genera and comprises more than 1300 species. They are widely cultivated and exploited by the local people in their geographical areas of distribution [7]. Plants of this family (including ginger, turmeric, galanga, etc.) have been widely used in traditional Chinese medicine and foods, especially in China and Southeast Asia. Galanga (also called galangal, galingale, or galingale) is a species of the ginger family. There are two different species of galanga: smaller galanga (*Alpinia officinarum* Hance) and greater galanga (*Alpinia galanga* Willd.). In southeastern Asian countries, greater galanga rhizomes are widely used as spices or as ginger substitutes. By contrast, smaller galanga is cultivated on a large scale in southern China, especially in Guangdong and Hainan provinces. Its rhizome is used as a spice when it is fresh and as a traditional medicine after the drying process [8].

Over the past two decades, scholarly inquiry on both has predominantly focused on phytochemical investigations [9,10]. Previous work showed that they have some similar chemical constituents, such as diarylheptanoids, flavonoids, essential oils, terpenes, phenylpropanoids, and glycosides, but the main chemical components differ. *A. galanga* is rich in a variety of phenolic compounds and essential oils, whereas *A. officinarum* is rich in flavonoids and diarylheptanoids [7]. In addition, investigations have shown that *A. galanga* and *A. officinarum* have many biological activities, including effectiveness as antitumor, antiviral, antimicrobial, antioxidant, antiallergic, and gastroprotective agents [7]. A significant amount of residue is often created after the extraction of phytochemical components from rhizomes; this residue is then usually disposed of or burned. However, there is a substantial amount of starch in these residues, which could be employed as an unusual starch source. To the best of our knowledge, there is almost no scientific information on the rhizome starch from galanga (smaller galanga and greater galanga).

On this basis, the objective of this study was to investigate the morphological, fine structure, pasting, and gelatinization properties and in vitro digestibility of starch from smaller galanga and greater galanga to provide basic information on the structure and physicochemical properties of galangal starch relevant to food industry applications. Another main target was to compare the physico-chemical characteristics of two galangas (*Alpinia officinarum* Hance and *Alpinia galanga* Willd) starches with a relatively well-studied ginger starch (*Zingiber officinale* starch). The results of this work do not only lay the foundation for future research on galangal starch but are also of great significance in promoting sustainable agriculture and broadening industrial applications of isolated starch from these sources.

## 2. Materials and Methods

### 2.1. Materials

The rhizomes of two galanga (smaller galanga and greater galanga) were purchased from traditional Chinese drug markets in Qinzhou City, Guangxi, China, in December 2021 and Gaozhou City, Guangdong, China, in December 2021, respectively. The samples of smaller galanga (*Alpinia officinarum* Hance) and greater galanga (*Alpinia galanga* Willd.) were authenticated by Dr. Kun Pan from Hainan Medical University (herbarium no. HNMU/2021/Pha/01 and HNMU/2021/Pha/02). The rhizomes of ginger (*Zingiber officinale*) were purchased from the local market (cultivated in Weifang City, Shandong Province, China) and transported fresh in bags to our laboratory.

The total starch assay kit and D-glucose assay kit (GOPOD) were purchased from Megazyme International Co., Ltd. (Wicklow, Ireland). The amylose content assay kit was purchased from Nanjing Jiangcheng Bioengineering Institute. (Nanjing, China). Pancreatin (Cat. No. P7545), amyloglucosidase (Cat. No. A7095), and proteases (Cat. No. P7012) were obtained from Sigma-Aldrich Company Ltd. (Shanghai, China). Other reagents were of analytical grade.

### 2.2. Isolation of Native Starches from the Rhizomes

The roots underwent washing, peeling, and chopping. Subsequently, they were homogenized with distilled water at 90 rpm/min for 3 min at room temperature. After two repetitions, the slurry was filtered using a 200-mesh nylon mesh. The filtrate was left in a beaker for 12 h, and the supernatant was discarded to collect the white precipitate. Yellow impurities were removed, and the precipitate was repeatedly washed with distilled water until the supernatant was clarified. After defatting the sample in hexane, it was centrifuged, and the precipitate was collected. An appropriate amount of sodium hydroxide solution (pH = 9.0) was then added, and the mixture was stirred at 25 °C for 6 h. Following stirring, the mixture was centrifuged again, and the precipitate was collected. The obtained starch cake was washed with distilled water and ethanol 4 to 5 times. The resulting precipitate was placed in a 45 °C air-blast drying oven and dried for 48 h. The starch was obtained after being crushed through a 160-mesh sieve and stored in a sealed bag at room temperature. The starch samples extracted from smaller galanga (*Alpinia officinarum* Hance), greater galanga (*Alpinia galanga* Willd.), and ginger (*Zingiber officinale*) were AOS, AGS, and ZOS, respectively.

### 2.3. Proximate Analysis and Amylose Content

The total starch and amylose contents were determined using the total starch determination kit (A148-1-1, Nanjing Jiancheng Bioengineering Institute, Nanjing, China) and amylose contents determination kit (A152-1-1, Nanjing Jiancheng Bioengineering Institute, Nanjing, China), respectively. Analyses were conducted for crude protein (AOAC method 984.13), moisture (AOAC method 930.15), ash (AOAC method 942.05), and crude fat (AOAC official method 960.39) [11].

### 2.4. Morphological Characterization

#### 2.4.1. Polarized Light Microscopy Analysis (PM)

Polarized light microscopy images were obtained by polarized light microscopy (DMBA400, Motic China Group Co., Ltd., Nanjing, China). A small amount of the starch sample was taken and mixed with 50% glycerin. A drop of this mixture was then placed on a glass slide, covered, and placed under a microscope. The polarized light mode was adjusted to observe the polarized cross of the starch granules [12]. 

#### 2.4.2. Scanning Electron Microscopy Analysis (SEM)

The morphology of starch granules was observed by a scanning electron microscope (Sigma 500, Carl Zeiss AG, Jena, Germany) at a 5 KV acceleration voltage [13].

#### 2.4.3. Confocal Laser Scanning Microscopy Analysis (CLSM)

The sample (1~2 mg) was stained with 3 μL of the APTS (8-amino-1, 3, 6-pyrene trisulfonic acid) solution and 3 μL of the sodium cyanoborohydride (1 mol/L) solution, placed at 30 °C for 15–18 h, washed five times with distilled water, and then suspended in 20 μL of 50% glycerol. A 1 μL sample was fixed on a glass plate containing a mixture of 2% agar and 85% glycerol, and the sample was observed by CLSM (LEICA TCS SP8, Leica Microsystems Ltd., Wetzlar, Germany) [12].

### 2.5. Granule Size Distribution

The particle size distribution was analyzed using a laser particle size analyzer (2000, Malvern Instruments Ltd., UK). Starch suspension with a mass fraction of 10% was prepared for the analysis. The refractive index of the particles was set at 1.52. The particle sizes corresponding to 10%, 50%, and 90% of the particles in the cumulative distribution were determined by the instrument and denoted as *d*_(0.1)_, *d*_(0.5)_, and *d*_(0.9)_, respectively. The volume mean diameter was D[3,4], and the surface mean diameter was D[2,3].

### 2.6. Multi-Scale Structure Analysis

#### 2.6.1. Crystalline Structure: X-ray Diffraction (XRD)

X-ray diffraction patterns of the samples were obtained using an X-ray diffractometer (XRD-7000, SHIMADZU, Kyoto, Japan). Scanning was performed at a rate of 2°/min at room temperature over a range from 10° to 40°. Relative crystallinity was calculated and analyzed using Jade 6.5.

#### 2.6.2. Short-Range Ordered Structure: Fourier-Transform Infrared Spectroscopy (FTIR) and Raman Spectroscopy

The starch samples were mixed with KBr powder and pressed into tablets. These samples were analyzed using FTIR and scanned from 400 cm^−1^ to 4000 cm^−1^ with a resolution of 4 cm^−1^ for 64 scans against an air background. Final spectral analysis was performed using Omnic 8.2 and Peakfit 4.0 software [14]. The Raman spectra of starch were tested by a laser Raman spectrometer (LabRAM HR Evolution; Horiba, Kyoto, Japan) and the samples were excited with a 785 nm laser source, and the spectra were scanned from 400 to 3200 cm^−1^ with a resolution of 4 cm^−1^ for 64 scans. Finally, the full width at half maxima (FWHM) at 480 cm^−1^ was calculated using Peak fit 4.0 software fitting to characterize the short-range ordered structure of starch molecules [14].

### 2.7. Molecular Structure: Size-Exclusion Chromatography (SEC)

According to the method of Li et al., [15] 10 mg of the sample was used and resuspended in 1.8 mL of water. Then, 0.2 mL of sodium acetate (pH 3.5), 10 mL of NaN_3_ (40 mg/mL), and 20 μL of isoamylase (1400 U) are added and mixed thoroughly. The mixture is incubated in a 37 °C water bath for 3 h. A total of 10 mL of anhydrous ethanol is added and centrifuged at 4000× *g* for 10 min. A total of 1 mL of DMSO/LiBr solution is added, and the sample dissolves at 350 rpm and 80 °C for 2 h using a mixer. The molecular structure of the entire branched starch can be analyzed using SEC (1100, Agilent, Santa Clara, CA, USA), with a flow rate of 0.6 mL/min of the DMSO/LiBr solution for elution. A GRAM-100 and GRAM-1000 column (Polymer Standards Service GmbH, Mainz, Germany) and a differential refractive index detector (Waters Corporation, Milford, MA, USA) were used. Using the Mark–Houwink equation, the elution time is converted to Rh (mean chain length) with a refractive index increment (*dn*/*dc*) value of 0.0717 mL/g and from Rh to the degree of polymerization (DP) X to obtain X(Rh).

### 2.8. Physicochemical Properties

#### 2.8.1. Solubility (SOL) and Swelling Power (SP)

The SOL and SP of the samples were determined according to the method reported by Li et al. [16] with slight modifications. In total, 400 mg of starch was mixed with 40 mL of distilled water in a centrifuge tube, and the sample was continuously shaken in a water bath at 55, 65, 75, 85, and 95 °C for 30 min, and then the centrifuge tube was cooled to room temperature. The tubes were cooled to room temperature and then centrifuged at 4000 r/min for 20 min. The supernatant was poured into an aluminum box that had been kept at a constant weight and dried at 100 °C until constant weight was achieved, and the precipitate in the centrifuge tube was weighed. The solubility and swelling power were calculated as follows:(1)$$\mathrm{SOL}\;(\%)=\frac{\mathrm{WS}\times 100}{W}$$
(2)$$\mathrm{SP}\;(\%)=\frac{\mathrm{WSG}\times 100}{W(100-\mathrm{SOL})}$$
WS, the mass of the dried supernatant; W, sample weight; and WSG, the mass of solubilized granules.

#### 2.8.2. Pasting Property

The starch sample (3 g) was placed in the aluminum container of an RVA instrument, and 25 mL of distilled water was added. The sample was thoroughly dispersed using a paddle and analyzed using a Rapid Visco Analyzer -TecMaster 15A03004 (PerkinElmer, Inc., Hagersten, Sweden) program: starting temperature was set at 50 °C for 1 min, then the temperature was increased to 95 °C at a rate of 12 °C/min and held for 2.5 min, followed by a temperature decrease to 50 °C at the same rate which was held for 1 min. Stirring was initiated at a speed of 960 rpm for 10 s and then reduced to 160 rpm [17].

#### 2.8.3. Thermal Property

Starch samples (3 mg) were placed in an aluminum crucible, and 7 μL of distilled water was added with a pipette gun, sealed liquid press, and placed in a refrigerator at 4 °C overnight to allow for full hydration [18]. Test conditions for the DSC (TA Instruments, New Castle, DE, USA) were as follows: temperature scan range 30–110 °C, a temperature increase rate of 10 °C/min, and nitrogen as a protective gas. An empty crucible was used as a reference, and the onset temperature (*To*), peak temperature (*Tp*), conclusion temperature (*Tc*), and enthalpy (Δ*H*) on the heat absorption curve were recorded.

#### 2.8.4. Rheological Properties of Starch Gels

The storage modulus (G′), loss modulus (G″), and loss tangent (tan δ) of the starch gels were analyzed using a rheometer (AR-G2, TA Instruments Ltd., New Castle, DE, USA) equipped with a 35 mm diameter parallel plate geometry [19]. Starch suspensions (10% *w*/*v*) were first gelatinized (95 °C, 50 min) and then transferred to the test plate. The sample volume was approximately 1.5 mL to fill the gap (1 mm) between the two plates completely. The samples were swiftly placed on the rheometer plate and preheated to 25 °C. Dynamic viscoelastic tests were performed at an angular frequency of 0.1–10 rad/s at 1% strain in the linear viscoelastic region. The storage modulus (G″) and loss modulus (G″) were recorded. Another portion of the sample suspension was transferred directly onto an aluminum plate and sealed with silicone oil (to prevent water evaporation) for dynamic temperature rheological characterization experiments with a heating range from 25 °C to 95 °C. Each sample underwent measurement three times to verify reproducibility.

### 2.9. In Vitro Digestibility

The in vitro digestibility of starch was determined according to the method reported by Xu et al. [20] and the starch digestion curves were fitted according to the first-order kinetic equation, and the simulated fitted starch digestion curves were constructed for further analysis.

We took 0.6 g of the sample and added it to 3 mL of distilled water in a 100 mL beaker, then placed it in a boiling water bath for 30 min. After cooling to 37 °C, we added 3 mL of pancreatic α-amylase (250 U) and 15 mL of pepsin (3200 U in HCl solution) to 100 mL of the sodium acetate buffer solution (0.5 mol/L, pH = 5.2), and introduced it into the gelatinized sample for enzymatic hydrolysis for 30 min. After the hydrolysis, we neutralized it with 15 mL of sodium hydroxide (0.02 M). Then, we added 75 mL of the sodium acetate solution (pH = 6), 15 mL of pancreatic protease (4 USP), and 15 mL of glucoamylase and continued the reaction. Subsequently, at 0, 10, 20, 30, 40, 50, 60, 90, 120, 150, and 180 min, we withdrew 600 μL of the digest from the centrifuge tube. The reaction was stopped by adding 1800 μL of anhydrous ethanol, and the glucose content in the supernatant was measured using the GOPOD reagent kit. We calculated the contents of rapidly digestible starch (RDS), slowly digestible starch (SDS), and resistant starch (RS) as follows:
RDS (%) = (G20 − G0) × 0.9 × 100/TS(3)
(4)SDS (%)=(G120−G20)× 0.9×100/TS
(5)RS (%)=(1−RDS−SDS)×100

G0, G20, and G120 represent the amount of glucose released at 0, 20, and 120 min, respectively, and TS is the total starch content.

### 2.10. Statistical Analysis

All the data reported were the average values of at least triplicate measurements. Statistical analysis of all experimental data was conducted using SPSS (version 16.0, Chicago, IL, USA), employing one-way ANOVA and Duncan’s multiple range tests for group comparisons. Graphical representations were created with Origin 2021a software (Origin Lab Corporation, Northampton, MA, USA).

## 3. Results and Discussion

### 3.1. Yield, Proximate Analysis and Amylose Content

The yields, chemical compositions, and amylose content of AOS, AGS, and ZOS are shown in Table 1. The yield of ZOS was 53.34% on a dry basis, with a purity of 97.83%, which was consistent with the results reported by Li et al. [21]. Compared to ZOS, the yields of AOS and AGS were significantly lower at 22.10 wt% and 15.73 wt%, respectively. It was reported that extraction yields of starch from various rhizomes ranged from 6.9 to 56% [21,22,23]. Typically, the extraction yield of starch is influenced by the harvesting season of rhizomes and the chosen extraction method [22,24]. Regarding purity, both AOS and AGS were comparable to ZOS, with values reaching as high as 97%. The obtained yield and purity for galanga starch indicate that these rhizomes hold significant potential as a starch source for commercial use.

The starches isolated from three rhizomes (AOS, AGS, and ZOS) contained 11.13–11.8% moisture, 0.86–0.94% protein, 2.08–2.72% lipids, and 0.06–0.09% ash, with no statistically significant differences observed (Table 1). The moisture content observed in this study aligns with the previously reported approximate value of 9–11% for starch from six Zingiberaceae species [25]. The protein and ash contents of the three rhizomes starches were similar (<0.10%). It is noteworthy that despite the use of hexane and ethanol for defatting during the extraction of rhizome starch, the lipid content in all samples remains higher compared to other starch types found in turmeric rhizomes [26]. This is primarily due to the naturally high lipid levels in plants of the Zingiberaceae family.

The amylose content significantly influences the properties of starch and governs the majority of its applications. As can be seen in Table 1, the amylose content of two galanga rhizome starches (24.14% and 18.81%) was significantly lower than that of ZOS (28.39%). The results fell within the previously reported range (12–34%) for starch from the rhizome of another Zingiberaceae species [25,27,28].

### 3.2. Morphological Characteristics and Granule Size Distribution

The morphology and size of the granules can significantly influence the physicochemical and rheological properties as well as the nutritional function of starch.

Herein, various morphological analytical techniques, including optical microscopy, confocal laser scanning microscopy, and scanning electron microscopy, were employed to investigate the granule characteristics of these rhizome starches. No discernible contaminants were observed (Figure 1(A1–A3)), confirming that the isolated starches had sufficiently high purity suitable for subsequent analysis. Interestingly, AOS and AGS exhibited an elongated shape, differing from commonly known round starch particles. The shape of ZOS granules was found to be elliptical, and triangular flake forms with diameters ranging from 2 to 16 μm were consistent with previous findings [21,27]. The starch granule sizes of AOS and AGS were 8–26 and 6–28 μm, respectively. The size and shape distribution of starch granules exhibited variations across different plant species, thereby influencing the functional properties of starch.

The phenomenon of birefringence, colloquially known as Maltese crosses, reflects the fact that starch granules are a type of crystal ball and the directionality of the internal crystal structure [29,30]. The granules of ZOS were dark at the center and presented the typical “Maltese cross”, whereas granules of AOS and AGS had different birefringence patterns, with the points of these crosses located at the end of two intersecting lines of starch particles. All granules of AOS and AGS (Figure 1(B1,B2)) exhibited strong birefringence, reflecting that there is a high degree of molecular orientation. However, in ZOS, most of the granules exhibited weak birefringence, whereas a few exhibited strong birefringence (Figure 1B3). This phenomenon may be attributed to the higher levels of amylose content and relative crystallinity content (Table 1) [31].

SEM images (Figure 1(C1–C3)) not only validated the findings of shape and size as observed through optical microscopy but also provided additional information regarding the topographical characteristics of the exposed surfaces of the starch granules. AOS and AGS present elongated shapes, while ZOS possess an elliptical shape and are arranged in a layered manner. According to existing research [13,27,32,33,34], some varieties of banana starch, as well as starches from cactus and Araceae plants, also exhibit a rod-like shape. For example, rod-shaped banana starch exhibits higher solubility compared to cassava, potato, and sorghum starches. Additionally, the viscosity characteristics of rod-shaped banana starch are significantly superior to those of corn starch; at the same concentration, the Brabender viscosity of banana starch paste is four times that of corn starch paste. Therefore, we can infer that these rod-shaped starches have higher solubility and enhanced gelatinization properties, which is further corroborated by subsequent studies [32,35]. Therefore, these natural rod-like starch particles could hold promise for various applications. Additionally, SEM results revealed smooth surfaces on the starch granules of AOS and ZOS, whereas AGS displayed rough surfaces (Figure 1(C1–C3)).

CLSM has enabled the study of the internal structure of starch granules, including growth rings, channels, pores, and the distribution of amylose [36]. After staining all samples with APTS for CLSM observations, the corresponding results are shown in Figure 1(D1–D3). Growth rings were clearly visualized in AOS, AGS, and ZOS. These results suggest the alternative arrangement of the crystalline and amorphous regions in all starch granules.

Figure 1(E1–E3) shows the representative particle size distributions (PSDs) of the three rhizome starches. The corresponding parameters were calculated and summarized in Appendix A. All starches showed bimodal size distributions, with the peak diameters being 18.2, 18.1, and 10.4 μm, respectively, in AOS, AGS, and ZOS (Figure 1(E1–E3)). AOS and AGS exhibited a dominant peak in granule size distribution (Peak I) within the range of ca. 5–50 μm, along with a smaller shoulder peak (Peak II) between ca. 0.6 and 5 μm. The granule size distribution peak of ZOS shifted towards significantly smaller size values (ca. 0.6–25 μm) compared to those of AOS and AGS. Consistently, ZOS displayed evidently smaller-sized parameters than AOS, as shown in Appendix A. These findings are in agreement with the results obtained from optical microscopy and SEM. In conclusion, the granule morphology exhibited similarities between AOS and AGS, but significant differences were found between these two and ZOS.

### 3.3. Multi-Scale Structure

#### 3.3.1. Crystalline Structure

X-ray diffraction is commonly used to reveal the presence and characteristics of starch crystal structures [37]. The X diffraction patterns of the three rhizome starches are shown in Figure 2A. AOS and ZOS show distinct strong diffraction peaks at 15°, 17°, and 23°, along with weak peaks at 5.6° and 18°, indicative of the C-crystalline mode [37], while AGS shows distinct peaks at 15°, 17°, 18°, and 23°, indicating an A-type crystal structure. This result aligns with previous reports indicating the occurrence of multiple crystal patterns within the same family [16,38]. Additionally, all these starches exhibited an apparent peak at approximately 20°, indicating the presence of V-type crystals formed by starch and endogenous lipids. The relative crystallinity of the three starches ranged from 28 to 35.26% (Table 2). It is evident that AGSs were of lower crystallinity than AOS and ZOS. This may be due to differences in the amylose content, molecular weight distribution, and types of crystallinities within the same species [39].

#### 3.3.2. Short-Range Ordered Structure

FTIR was used to analyze differences in the starch helical structure. In all the starches in the spectra (Figure 2B,C), an extremely broad band appeared at 3450 cm^−1^ as a stretching vibrational peak of the free hydroxyl group (-OH); the peaks at 1160 cm^−1^ and 1080 cm^−1^ were related to the C-O stretching vibrational peaks of C-O-H; and 929 cm^−1^ was the asymmetric stretching vibrational peak of the D-glucopyranose ring [40,41]. These peaks are typical starch absorption peaks. However, no new absorption peaks appeared, indicating that the three starches of the same family are chemically bonded in the same way. In these peak plots, it is not difficult to find that there are clear differences in the intensities exhibited by the three starch types in these peaks, which may be caused by the fact that the three starches contain different amounts of these groups as well as the different crystallinities of the starch. In the spectrograms, the 1047 cm^−1^, 1022 cm^−1^, and 995 cm^−1^ bands are commonly used to reflect the ordered and double helical structure of starch. Therefore, R_1047/1022_ and R_995/1022_ were used to assess the degree of order and the amorphous content of starch [42]. The R_1047/1022_ and R_995/1022_ of AOS were higher than those of AGS and ZOS (Table 2). It indicates that AOS has more ordered and double helical structures among the three starch types, which is a result that is consistent with the previous crystallinity results.

The short-range ordering degree of starch can further determine the use of Raman spectroscopy detection. The Raman spectra of the three starches are presented in Figure 2A. The positions of the characteristic group peaks are consistent, indicating that the three starch types share the same chemical composition unit. The bands at 3490 cm^−1^, 2920 cm^−1^, and 1400 cm^−1^ in the spectra are typically associated with hydroxyl (-OH) stretching vibrations, C-H bond stretching vibrations, and C-H bond bending vibrations or specific C-O vibrations, respectively [43]. The differences in the intensities of these three starches in these bands may be related to the hydrogen bond formation and crystallinity of the starch molecules, indicating significant structure differences among them. The peak at 480 cm^−1^ in the spectrum corresponds to the vibration of the carbon skeleton (α-glucose ring). The half peak width (FWHM) of this peak can be used to estimate the variation in the short-range ordering of the starch [42]. The smaller the width of the half peak at 480 cm^−1^, the more ordered the starch structure is. In Table 2, it can be observed that the FWHM values of the three starches are AOS, AGS, and ZOS in ascending order, and it can be shown that AOS is more molecularly ordered than the other two starches. This result is consistent with the FT-IR results.

#### 3.3.3. SEC

Figure 2F shows the SEC weight chain length distribution (CLD), W(logX) versus DP (X) for each chain of debranched starch. Table 3 calculates the molecular size and distribution for both amylose and amylopectin, the degree of polymerization (DP), and amylose content (AM_SEC_). The distribution of low, middle, and high molecular weight peaks was observed. The first peak (DP 3~36) represented branched short amylopectin (AP1), spanning only one crystal lamellar layer. The second peak, dp 36–100, corresponded to longer amylopectin branching (AP2), spanning multiple lamellar layers. The third peak, at dp 100–15,000 of the SEC weight distribution, represented the amylopectin peak (AM) [15]. The weight-based proportion of the different chain components is determined by the area proportion of the corresponding component in the total area. Distinct differences in AP2 and AM peaks towards shorter chain lengths are notable in ZOS compared to AOS and AGS, which is consistent with ZOS’s higher amylose content, as detailed in Table 3. This higher amylose content, suggesting tighter gel formation during heating, limits starch granule swelling and water absorption, reducing peak viscosity. In contrast, the amylose average chain length and short amylopectin chains of AOS are longer than those in ZOS and AGS despite AOS not having the highest amylose content. This indicates that AOS starch, with higher crystallinity and a more ordered structure, as reflected in the DP and polymerization degree, might influence its functional properties, as shown by minimal differences in the AP1 peak across all three starches. Previously, it was found that when measuring chain length distribution using the SEC technique, caution is needed due to potential errors in the calibration process and the Mark–Houwink formula, resulting in a lack of accuracy in the detection of short-amylopectin starch [44].

### 3.4. Physicochemical Properties

#### 3.4.1. Swelling Power (SP) and Solubility (SOL)

Generally, the SOL and SP of starch reflect the solubility and water-holding capacity of swollen starch during starch heating, respectively [45]. Figure 3A,B show the SOL and SP of the three starches at 55–95 °C at 10 °C intervals. The SOL and SP of all the starch types increased with temperature, peaking at 95 °C. At the lower temperature (55 °C), the starches had not yet begun to pasteurize, resulting in both SOL and SP being low. The SOL and SP of AOS increased rapidly up to 65 °C and continued to rise with further temperature increase. In contrast, AGS and ZOS exhibited a significant increase in SOL and SP only after reaching 75 °C. This difference may be attributed to weaker hydrogen bonding between the AOS double helix compared to the other two starches [46]. Meanwhile, with increasing temperature, the SOL and SP of the three starches showed varying rates of increase: AOS > AGS > ZOS. This difference is mainly related to the starch’s crystalline structure and the length of the amylopectin chains. A higher content of the crystalline zones and shorter amylopectin chain lengths restrict the hydration and solubility of the starch [17,47].

#### 3.4.2. Pasting Property

The RVA curves and corresponding pasting parameters of these starch samples are shown in Figure 3C and Table 4. AGS starch exhibits the highest peak viscosity through viscosity, breakdown viscosity, final viscosity, and setback viscosity. AOS starch has the lowest pasting temperature, which is consistent with the gelatinization temperature shown by DSC (Table 5). AOS and AGS had an overall significantly higher pasting viscosity than ZOS. A previous study indicated that starch had a higher amount of medium, and long amylopectin branches exhibited a higher viscosity. The higher content of amylose in ZOS led to the formation of a tighter gel structure during heating and water absorption, which limited the swelling of the starch granules and the further absorption of water, thus reducing the peak viscosity [48], whereas the lower trough viscosity of ZOS and AOS may be due to the fact that starch with a higher content of amylopectin is more likely to form crystals during cooling and retrogradation, which may limit inter-particle movement to a certain extent and reduce the fluidity of the paste, thus lowering the bottom of the trough viscosity [49]. The through viscosity rapidly increases after the temperature decreases, resulting in a sharp peak. This phenomenon, observed in previous studies [50], is likely due to the rapid reorganization of starch molecules. The setback viscosity is a property of increased viscosity or resistance to flow when the sample is cooled after gelatinization by heating and is used to evaluate the gelation characteristics and retrogradation ability of starch [51]. The ZOS and AOS showed minimal setback viscosity, indicating a high resistance to regrowth. This property is associated with its smaller starch granule size and higher content of amylose. A possible explanation is that the special structure of amylopectin somehow interferes with or limits the interactions between the amylose, which leads to an expected drop in setback viscosity [52].

#### 3.4.3. Thermal Property

The thermal properties of the three starches were analyzed by DSC, and the gelatinization parameters are shown in Table 5. *To*, *Tp*, *Tc*, Δ*T* and Δ*H* in all samples ranged as follows: 61.53–73.15 °C, 69.42–82.92 °C, 91.48–94.72 °C, 21.57–29.95 °C and 20.25–22.06 J/g, respectively. AGSs hold the highest onset and peak temperatures. The differences in the conclusion temperatures and enthalpies of pasting for the three starches were not significant, and it is known that the ending temperatures and enthalpies of pasting were similar for the three starches. The thermal properties of starch are affected by various factors, such as starch granule size, amylose content, etc. [53]. The onset of the pasting temperature of ZOS was higher than that of AOS, probably because of the negative correlation between starch granule size and pasting temperature [54]. Meanwhile, the amylose content of starch and the distribution of amylose chain length also has an effect on the pasting temperature [55]. The more the starch granules contain amylose content, the higher the swelling resistance, and the higher temperature is needed to leach out the amylose molecules and achieve the maximum swelling; therefore, the straight-chain content is positively correlated with the pasting temperature, and the amylose chain length is positively correlated with the pasting temperature, which also explains that the pasting onset and peak temperatures of ZOS are higher than those of AOS, and this is basically consistent with the trend of pasting characteristics. However, on the other hand, the results of DSC and pasting properties do not fully correspond. This discrepancy may be because RVA tests are conducted under higher humidity and shear conditions, affecting starch solubility and gelatinization, whereas DSC measures the thermodynamic properties of starch gelatinization. Due to the different sensitivities of these methods, RVA reflects viscosity changes while DSC primarily reflects heat changes, causing the same starch type to exhibit different behaviors in each test.

#### 3.4.4. Rheological Properties

Figure 4A–C show the changes in storage modulus (G′), loss modulus (G″), and loss factor (tan δ) with the increasing angular frequency for the three starches at 25 °C. Both G′ and G″ of the three starch gels increase with the increase in angular frequency, and G′ is always larger than G″, but the AOS starts G′ decreasing after the angular frequency reaches a certain value. This indicates that the elastic properties of the three starches are stronger than the viscous properties, showing a weak gelation behavior. Apparently, the G′ values of ZOS are significantly higher than those of AOS, which indicates that its starch is more viscoelastic and the gel network structure is stronger. The storage modulus of AOS decreases at a certain point with increasing angular frequency, which may be due to the fact that the high frequency causes the molecular network of this starch to undergo excessive strains, which exceeds its self-recovery ability and leads to the partial collapse of the network structure [56]. This may be due to differences in the molecular structure and cross-linking density within the starch networks that prevent other starch networks from collapsing. The loss factor (tan δ) is the ratio of G′ and G″, which reflects the relative viscosity or elasticity of the viscoelastic samples. tan δ values greater than one indicate that the samples are more viscous than elastic, and they are viscous fluids; the more mobile they are, the more values less than one indicate that the samples are more elastic than viscous, and they are sols or gels. The tan δ values of all the starch types are less than one, indicating that all the starches are elastic, and the gels can maintain a stable shape and structure. The loss factor of AGS rises rapidly when the angular frequency rises, quickly exceeding 0.1, and then decreases at high frequencies. Both AOS and ZOS are less than 0.1 at low frequencies, which indicates that the AOS and ZOS have good elasticity and structural stability [57].

The temperature scanning rheological curves of the three starch gels are shown in Figure 4D,E. The G′ values of all starch gels are larger than G″, and the tan δ values are less than one, indicating that the elastic behavior is larger than the viscous behavior of all samples. As the temperature rises, all the starch gels show similar trends, with the energy storage modulus and loss modulus showing a decreasing trend from the beginning of the temperature rise, followed by stability. It indicates that as the temperature increases, starch granules gradually expand, the crystalline regions remaining in the starch granules melt, the starch granules deform, rupture, and disintegrate, and the molecular mobility increases, leading to the weakening of interchain interactions, which, in turn, leads to a gradual decrease in G′ and G″ [58]. Under the same temperature conditions, the G′ and G″ of AOS and ZOS were significantly higher than those of AGS, indicating that these two starch paste systems have more entanglement points between molecular chains within the gel system and the network structure of the gel system is stronger, i.e., it has a stronger three-dimensional network structure.

### 3.5. In Vitro Digestibility

Starch can be classified into rapidly digestible starch (RDS), slowly digestible starch (SDS), and resistant starch (RS) by the rate and extent to which the starch is digested under in vitro conditions. RDS is rapidly digested and absorbed in the small intestine, which may lead to a rapid rise in blood glucose levels, which is detrimental to glycaemic control in diabetic patients [59]. Comparatively, SDS is digested more slowly in the small intestine, with a moderate rise in blood glucose, and is considered a more desirable type of dietary starch [60]. RS is not digested in the small intestine, and after entering the large intestine, it can be fermented by gut microorganisms to produce short-chain fatty acids, among other things, which can help to regulate blood glucose, lower cholesterol, and prevent obesity, colon cancer and gallstones [59]. Therefore, foods high in resistant starch have similar health benefits to dietary fiber and can be promoted as healthful food components. The RDS, SDS, and RS of the three starches as a percentage of the total starch are shown in Table 5. Also, based on the LOS model, the digestion profiles of the starch can be evaluated as a comparison of the digestion differences (Figure 3D–F) and the values of the digestion rate (*k*) and digestion endpoint (*C*_∞_) obtained by fitting the first-order kinetic equation to the in vitro starch digestion results using the LOS model are listed (Table 6). It is easy to find that the RDS, SDS, RS, and C_∞_ values of the three starch types are not significantly different, but the standard deviations of the individual values of AGS and ZOS are relatively large, suggesting that these two starches have large repetition errors, which may affect the accuracy of the results. Due to the properties of AOS starch, such as higher crystallinity, larger particle size, and a more ordered structure, it is speculated that it may have more potential for high-resistant starch content [19,59].

## 4. Conclusions

The results of this study show that the three different ginger family rhizome starches have the potential to be non-traditional sources of starch, which is characterized by their elongated shape. Meanwhile, the differences in total starch content and proximate analyses of the three starches were not significant. However, the extraction rate of ZOS was much higher than the other two starches in terms of extraction rate. There was not much difference in the morphology of the three starches, but there was a difference in the particle size. According to the analysis of XRD, FTIR, and Raman spectra, AOS and ZOS belong to the C-type crystal structure, and AGS belongs to the A-type crystal structure; meanwhile, there are clear differences in the crystallinity and orderliness of the three starches. The differences in starch structures led to significant differences in the physicochemical properties of the three starches, including the SP, SOL, thermal properties, pasting properties, and rheological properties of the three starches. These findings provide a new research basis for the rhizome starches of Zingiberaceae, as well as an important guide for the development of Zingiberaceae starch resources and their use in food processing.

## Figures and Tables

**Figure 1 foods-13-01784-f001:**
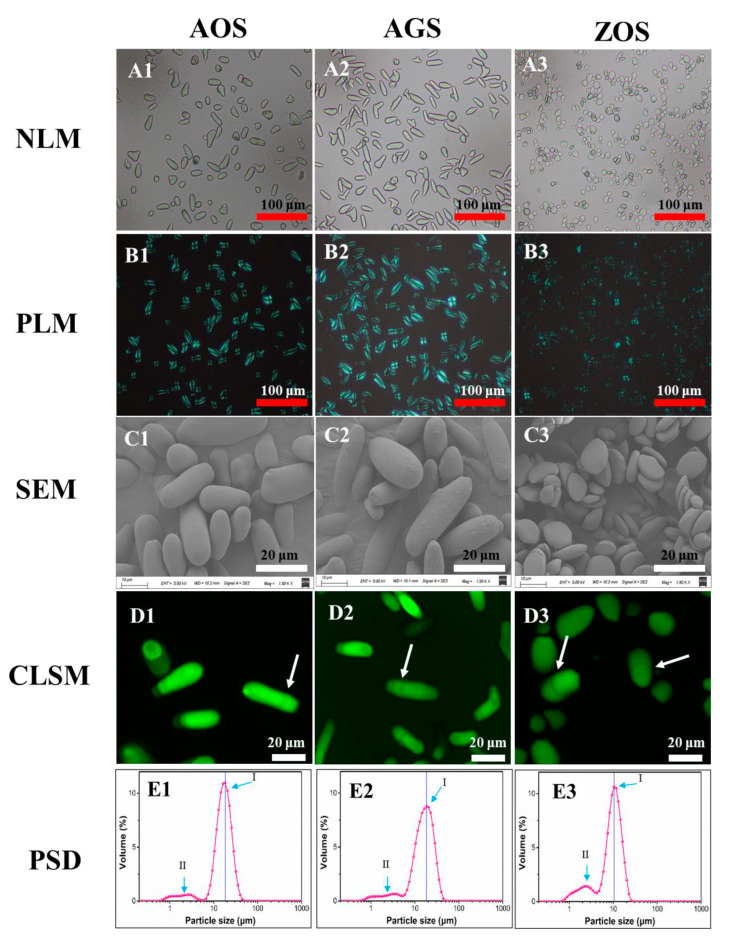
Microstructure of the three rhizome starches under optical microscopy (**A1**–**A3**), polarized light microscopy (**B1**–**B3**), scanning electron microscopy (**C1**–**C3**), confocal laser scanning microscopy (**D1**–**D3**), and particle size distributions (**E1**–**E3**). Red scale bars: 100 μm; white scale bars: 20 μm.

**Figure 2 foods-13-01784-f002:**
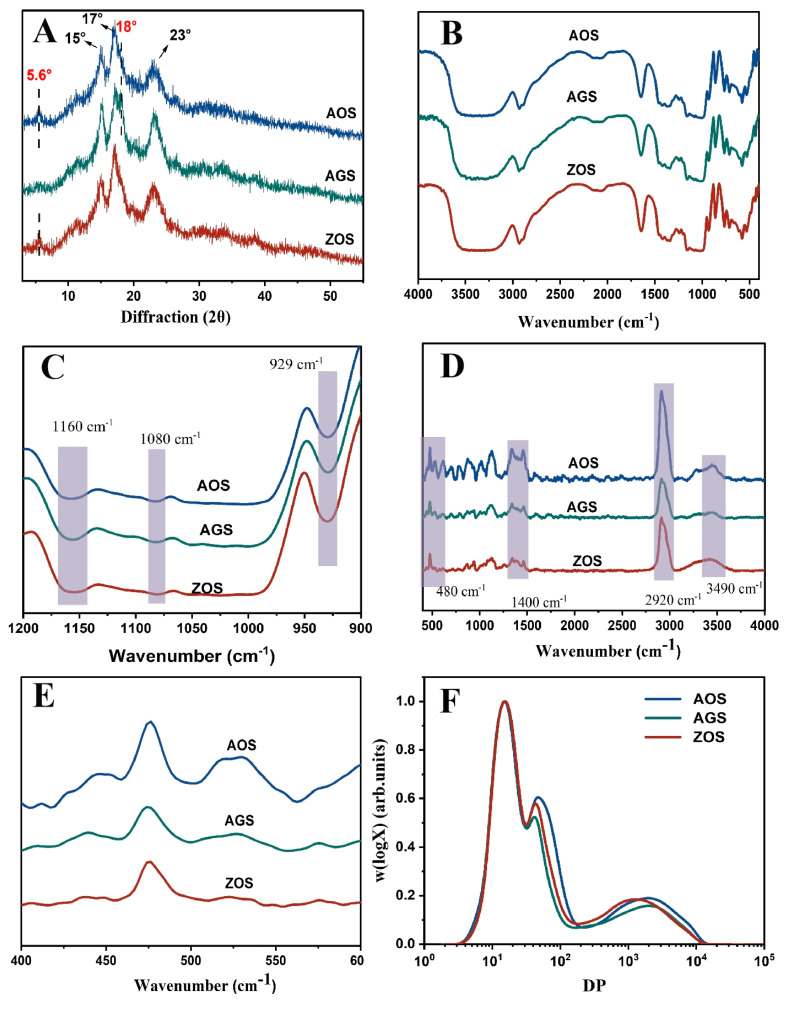
XRD (**A**), FTIR (**B**,**C**), Raman (**D**,**E**) and chain length distribution (**F**) of the three rhizome starches.

**Figure 3 foods-13-01784-f003:**
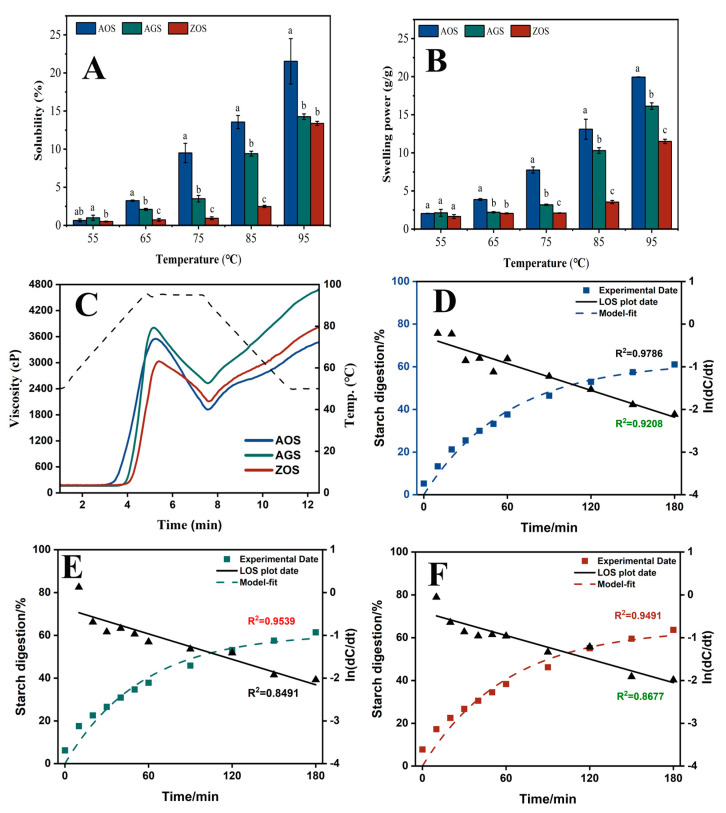
Solubility (**A**), swelling power (**B**) and RVA curves (**C**), digestion curves, model fit curves, and LOS plots ((**D**), AOS; (**E**), AGS; (**F**), ZOS) for the three rhizome starches. Different characters (a–c) on the top of columns represent significant differences between the AOS, AGS, and ZOS (at the same temperature), *p* < 0.05 level.

**Figure 4 foods-13-01784-f004:**
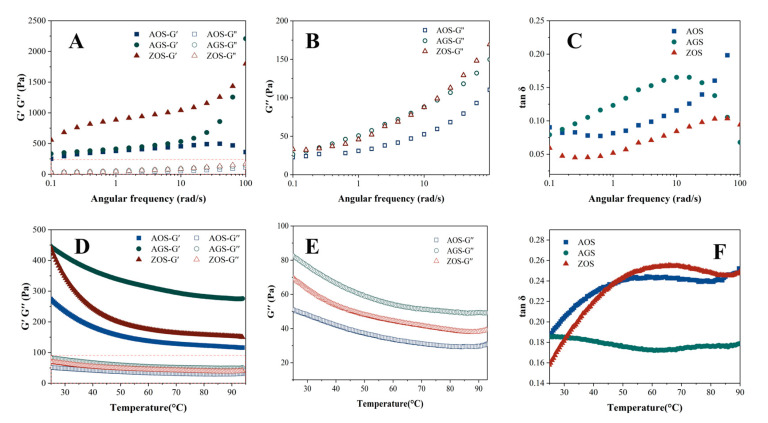
Rheological properties of three rhizome starches (**A**–**C**) frequency scan; (**D**–**F**) temperature scan).

**Table 1 foods-13-01784-t001:** Extraction yield, total starch, proximate analysis and amylose content of the samples (g/100 g dry sample).

Samples	Yield	Total Starch	Moisture	Protein	Lipids	Ash	Amylose Content
AOS	22.10 ± 3.15 ^b^	97.02 ± 1.09 ^a^	11.53 ± 0.63 ^a^	0.86 ± 0.02 ^a^	2.08 ± 0.29 ^a^	0.07 ± 0.01 ^a^	24.14 ± 0.73 ^b^
AGS	15.73 ± 2.03 ^b^	97.12 ± 1.74 ^a^	11.13 ± 0.60 ^a^	0.89 ± 0.01 ^a^	2.52 ± 0.09 ^a^	0.09 ± 0.02 ^a^	18.81 ± 0.96 ^c^
ZOS	53.34 ± 6.45 ^a^	97.83 ± 0.38 ^a^	11.88 ± 0.14 ^a^	0.94 ± 0.03 ^a^	2.72 ± 0.21 ^a^	0.06 ± 0.01 ^a^	28.39 ± 1.66 ^a^

Average values of three replicates followed by the standard deviation. The same column with different letters indicates significant differences (*p* < 0.05).

**Table 2 foods-13-01784-t002:** Absorbance ratios of 995/1022 cm^−1^ (R995/1022) and 1047/1022 cm^−1^ (R1047/1022), relative crystallinity (RC) and full width at half maxima (FWHM) of the samples.

Samples	RC	R_1047/1022_	R_995/1022_	FWHM
AOS	35.26 ± 1.02 ^a^	1.24 ± 0.02 ^a^	1.20 ± 0.00 ^a^	15.44 ± 0.35 ^c^
AGS	28.00 ± 1.44 ^b^	1.13 ± 0.00 ^b^	1.08 ± 0.00 ^b^	16.30 ± 0.10 ^b^
ZOS	32.62 ± 2.83 ^a^	0.99 ± 0.00 ^c^	0.99 ± 0.00 ^c^	16.99 ± 0.21 ^a^

Average values of three replicates followed by the standard deviation. The same column with different letters indicates significant differences (*p* < 0.05).

**Table 3 foods-13-01784-t003:** Chain lengths of the three rhizome starches.

Samples	AM_SEC_ (%)	Rh_AM_ (nm)	Rh_AP1_ (nm)	Rh_AP2_ (nm)	X_AM_ (DP)	X_AP1_ (DP)	X_AP2_ (DP)
AOS	22.44 ±0.35 ^b^	16.78 ± 0.22 ^a^	2.65 ± 0.00 ^a^	1.05 ± 0.00 ^a^	1419.38 ± 31.28 ^a^	58.96 ± 0.13 ^a^	13.39 ± 0.17 ^b^
AGS	21.63 ± 0.04 ^b^	15.09 ± 0.08 ^b^	2.51 ± 0.00 ^c^	1.05 ± 0.00 ^a^	1195.43 ± 22.66 ^b^	52.97 ± 0.07 ^c^	14.04 ± 0.16 ^a^
ZOS	23.79 ± 0.27 ^a^	14.16 ± 0.09 ^c^	2.56 ± 0.01 ^b^	1.05 ± 0.00 ^a^	1069.55 ± 12.57 ^c^	54.69 ± 0.26 ^b^	13.91 ± 0.21 ^ab^

Average values of three replicates followed by the standard deviation. The same column with different letters indicates significant differences (*p* < 0.05). AMSEC straight-chain starch content; AP1 short-branched amylose chain content; AP2 long-branched amylose chain content; Rh mean chain length; and X degree of polymerization DP.

**Table 4 foods-13-01784-t004:** Pasting properties of the three rhizome starches.

Samples	Pasting Temp (P_Temp_, °C)	Peak Viscosity (PV, mPa s)	Breakdown Viscosity (TV, mPa s)	Through Viscosity (TV, mPa s)	Final Viscosity (FV, mPa s)	Setback Viscosity (SV, mPa s)	Peak Time (P_Time_, min)
AOS	77.95 ± 1.13 ^b^	3363.00 ± 267.29 ^ab^	1978.00 ± 82.02 ^b^	1385.00 ± 349.31 ^a^	3450.00 ± 25.46 ^c^	1472.00 ± 107.48 ^b^	5.27 ± 0.00 ^ab^
AGS	78.28 ± 7.39 ^b^	3983.00 ± 251.73 ^a^	2511.00 ± 25.46 ^a^	1472.00 ± 277.19 ^a^	4707.50 ± 33.23 ^a^	2196.50 ± 58.69 ^a^	5.07 ± 0.09 ^b^
ZOS	86.38 ± 0.46 ^a^	3047.00 ± 18.38 ^b^	2190.50 ± 115.26 ^b^	856.50 ± 96.87 ^a^	3872.50 ± 78.49 ^b^	1682.00 ± 36.77 ^b^	5.47 ± 0.09 ^a^

Average values of three replicates followed by the standard deviation. The same column with different letters indicates significant differences (*p* < 0.05).

**Table 5 foods-13-01784-t005:** Gelatinization measured the three rhizome starches.

Samples	*T_O_* (°C)	*T_P_* (°C)	*T_C_* (°C)	Δ*T* (°C)	Δ*H* (J/g)
AOS	61.53 ± 0.03 ^c^	69.42 ± 0.59 ^b^	91.48 ± 0.37 ^a^	29.95 ± 0.35 ^a^	21.05 ± 3.25 ^a^
AGS	73.15 ± 0.46 ^a^	82.44 ± 0.59 ^a^	94.72 ± 2.75 ^a^	21.57 ± 3.06 ^b^	20.25 ± 0.67 ^a^
ZOS	65.49 ± 0.73 ^b^	82.92 ± 0.35 ^a^	93.35 ± 0.05 ^a^	27.86 ± 0.68 ^a^	22.06 ± 0.01 ^a^

Average values of three replicates followed by the standard deviation. The same column with different letters indicates significant differences (*p* < 0.05). To, gelatinization onset temperature; *Tp*, gelatinization peak temperature; *Tc*, gelatinization conclusion temperature; Δ*T*, gelatinization temperature range (*Tc* − *Tp*); and Δ*H*, gelatinization enthalpy.

**Table 6 foods-13-01784-t006:** In vitro digestive property parameters of the three rhizome starches.

Samples	RDS (%)	SDS (%)	RS (%)	C_∞_ (%)	*k*/10^−2^ (min^−1^)
AOS	15.97 ± 0.48 ^a^	31.71 ± 1.19 ^a^	49.34 ± 0.71 ^a^	62.41 ± 2.4 ^a^	0.01 ± 0.01 ^a^
AGS	16.37 ± 0.70 ^a^	30.60 ± 6.41 ^a^	50.86 ± 7.11 ^a^	61.16 ± 6.99 ^a^	0.01 ± 0.01 ^a^
ZOS	14.74 ± 2.29 ^a^	32.62 ± 5.54 ^a^	49.76 ± 7.84 ^a^	64.37 ± 9.01 ^a^	0.01 ± 0.01 ^a^

Average values of three replicates followed by the standard deviation. The same column with different letters indicates significant differences (*p* < 0.05).

## Data Availability

The original contributions presented in the study are included in the article/Appendix A, further inquiries can be directed to the corresponding authors.

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
