# Peer review of "Rod-Shaped Starch from Galanga: Physicochemical Properties, Fine Structure and In Vitro Digestibility"

_foods, 2024, doi:10.3390/foods13111784_

Round 1

Reviewer 1 Report

Comments and Suggestions for Authors

Generally, the manuscript foods-2993381 “Novel Rod-shaped Starches from Galanga: Physicochemical Properties, Fine Structure and In vitro digestibilityis well written and the subject is of scientific interest. Nevertheless, different sections in "materials and methods" need additions/corrections/clarifications. Moreover, the section "results and discussion" must be rewritten in a concise yet comprehensive manner, avoiding repetitions.

More detailed concerns that need to be addressed, are raised in the comments placed next to the text marked in yellow in the PDF document. 

Comments on the Quality of English Language

Moderate editing of English language required

Reviewer 2 Report

Comments and Suggestions for Authors

line 88 State how many extractions were done for the starch for true replications.

line 103 State the specific method numbers.

line 127  Why are both of these labeled D and what do the numbers in parentheses mean here?

lines 148 to 151  This statement is not clear.  Wouldn't the flow rate always be 0.6?

line 160 to 161 Minimize repetitive wording.

line 165 I don't think you mean solubilized, do you mean swollen starch?

line 223 Higher lipid contents compared to what?

line 241 No not None.

line 246 to 247  How were these numbers determined?

line 251  Reference to a crystal ball seems strange (better wording is preferred) and what is meant by directionality of the internal crystal structure?

line 258  A citation is needed for this statement.

line 271 to 272  I cannot differentiate smooth from rough in the figure.  Maybe larger figures are needed.

line 276  Its hard to see growth rings.  Maybe use arrows to point them out and a larger figure.

line 300 to 301  You need a citation for this statement.

line 307 You have not yet discussed enthalpies so either place the DSC information ahead of this section or do not mention it here.

line 388 paste not pasteurize

line 389 to 392  Did you do stats comparing across temperatures within samples type or did you compare the sample types within temperature for the stats in this graph?  This will change how you can state differences in the text.

line 413 to 415  High amylose is associated with greater setback and greater chance for retrogradation not lower.  Do you have breakdown data for Table 4.  

line 425 Delete peak temperature as it wasn't different stats wise.

line 429 to 431  AGS also had higher PT so how do you explain it being similar to ZOS?  

Figure 3  You need to explain the stats letters for 3A and 3B.

line 453 to 454 ZOS G' is also not much different than AGS so it is also higher than AOS.  How do you explain this?  Why would the other starches not partially collapse also?

line 467 How does the fact that the tan is lower than 1 indicate decreasing gel strengths?  

Figure 4  There needs to be different color lines rather than different symbols to make the figures clearer.  Also need figures and legends to be larger.  4A should only say G' on the y-axis correct?

line 508 to 510  I am not sure you can say this based on your data in Table 6.

line 517  There were differences in amylose content, so reword sentence.

Comments on the Quality of English Language

Needs minor grammatical corrections.

Reviewer 3 Report

Comments and Suggestions for Authors

L 2: Take „novel“ out. This word is generic.

L 12: If wt% pls. use wt% instead of %. Pls. check that elsewhere too.

L 17-18: What water content?

L 20: What is RS and AOS?

L 33: Viscosity in water?

L 72-79: Provide pictures of the samples or similar samples. Does harvest time and growing conditions have an influence on the starch. If not known pls. write that this was not investigated.

Fig. 1, SEM, scaling strips not well visible, especially not the number.

Table 4: At temperature these values are measured? Pls. write in the caption.

General: A more intensive comparison of own values to published values of other starches could be of interest to readers.

Reviewer 4 Report

Comments and Suggestions for Authors

The “Novel Rod-shaped Starches from Galanga: Physicochemical Properties, Fine Structure and In vitro digestibility” were broadly studied galanga starch as a novel starch source. This galanga starch showed unique thermal properties and high RS + SDS content.

For publication, please discuss below point.

1.     ZOS starch has been distinguished thermal properties specially pasting properties which show very low peak, trough, final, and setback viscosity compared to two others. Figure 3C, the pattern of pasting properties is unique. It seems that not finished pasting. Usually, final viscosity is lower than peak viscosity, but this is shown that higher final viscosity than peak viscosity. However, DSC study, AOS is lowest onset, peak and conclusion temperatures and ZOS sample showed that lower onset temperature than AGS. Please discuss about these thermal properties. Why between DSC and pasting properties, the data not inconsistent each other.

2.     Is there any other starch showing rod-shape? If so, please put on introduction.  And if there are other rod-shape starch there, is there any unique property of it. If there any, please put on the introduction.

3.     Table 4: pasting properties: please put the oneset & peak temperature.

Comments on the Quality of English Language

The “Novel Rod-shaped Starches from Galanga: Physicochemical Properties, Fine Structure and In vitro digestibility” were broadly studied galanga starch as a novel starch source. This galanga starch showed unique thermal properties and high RS + SDS content.

For publication, please discuss below point.

1.     ZOS starch has been distinguished thermal properties specially pasting properties which show very low peak, trough, final, and setback viscosity compared to two others. Figure 3C, the pattern of pasting properties is unique. It seems that not finished pasting. Usually, final viscosity is lower than peak viscosity, but this is shown that higher final viscosity than peak viscosity. However, DSC study, AOS is lowest onset, peak and conclusion temperatures and ZOS sample showed that lower onset temperature than AGS. Please discuss about these thermal properties. Why between DSC and pasting properties, the data not inconsistent each other.

2.     Is there any other starch showing rod-shape? If so, please put on introduction.  And if there are other rod-shape starch there, is there any unique property of it. If there any, please put on the introduction.

3.     Table 4: pasting properties: please put the oneset & peak temperature.

Round 2

Reviewer 4 Report

Comments and Suggestions for Authors

The article “Rod-shaped Starches from Galanga: Physicochemical Properties, Fine Structure and In vitro digestibility” has been modified by reviewer’s comments and I feel that is has been improved the clarity and quality of content. However, I still observe the scientifically unsoundness content and not enough of discussion.

1.      L 105 mentioned that “The total starch and amylose contents were determined using total starch determintion kit and amylose contents determination kit respectively”

Please put the manufacturer information of ‘kit’

Also, if you use the kit for those analysis Table 1 need to be replace the ‘apparent amylose content’ to ‘amylose content’

2.      L289-290 “superior gelatinization and textural properties” what is meat of “superior” this is not a scientific word. Please write the distinguishable properties of gelatinization and textural properties if rod-like shape starch contained distinguishable properties than other starch.  Also, this “superior gelatinization and textural properties” observed in AOS, ZOS, and AGS, whether or not, you need to significant discuss about it.

3.      L291 – 292 ‘elongated starch granules in the rhizomes of plants from the Rubiaceae family is a new 291 discovery’ à this is not making a sense at all. Need to clearness.

4.      Table 4. Pasting temp need to position at first of left side.

5.      Fig 3C. trough showed very unique patten than compared other starch à need to explanation and discussion about it.

6.      Table 5. DSC study showed that the starch observed very normal gelatinization temperature (onset, peak and conclusion) but Fig 3C showed that the final viscosity is much higher than peak viscosity and, in my knowledge, this is very unique pattern and need to be explanation.

Usually results of DSC and pasting properties are corresponding but your data seems very unique. I recommend discuss about it.  

Comments on the Quality of English Language

The moderate modification of English language needed. 
